

# Hierarchical multi-label classification model for science and technology news based on heterogeneous graph semantic enhancement

Quan Cheng[1,2], Jingyi Cheng[1], Jian Chen[2,3] and Shaojun Liu[2,3]

[1] School of Economics and Management, Fuzhou University, Fuzhou, Fujian, China
[2] Fujian Key Laboratory of Information Network, Fuzhou, Fujian, China
[3] Fujian Institute of Scientific and Technological Information, Fuzhou, Fujian, China

Corresponding author
Quan Cheng, chengquan@fzu.edu.cn

## ABSTRACT

In the context of high-quality economic development, technological innovation has emerged as a fundamental driver of socio-economic progress. The consequent proliferation of science and technology news, which acts as a vital medium for disseminating technological advancements and policy changes, has attracted considerable attention from technology management agencies and innovation organizations. Nevertheless, online science and technology news has historically exhibited characteristics such as limited scale, disorderliness, and multi-dimensionality, which is extremely inconvenient for users of deep application. While single-label classification techniques can effectively categorize textual information, they face challenges in leading science and technology news classification due to a lack of a hierarchical knowledge framework and insufficient capacity to reveal knowledge integration features. This study proposes a hierarchical multi-label classification model for science and technology news, enhanced by heterogeneous graph semantics. The model captures multi-dimensional themes and hierarchical structural features within science and technology news through a hierarchical transmission module. It integrates graph convolutional networks to extract node information and hierarchical relationships from heterogeneous graphs, while also incorporating prior knowledge from domain knowledge graphs to address data scarcity. This approach enhances the understanding and classification capabilities of the semantics of science and technology news. Experimental results demonstrate that the model achieves precision, recall, and F1 scores of 84.21%, 88.89%, and 86.49%, respectively, significantly surpassing baseline models. This research presents an innovative solution for hierarchical multi-label classification tasks, demonstrating significant application potential in addressing data scarcity and complex thematic classification challenges.

# INTRODUCTION

In the context of the new era, technological innovation has emerged as a fundamental driver of social development. As technological achievements and advancements proliferate, science and technology news has increasingly become a vital medium for disseminating such information (*Zhu & Wang, 2020*). However, the field confronts the challenge of information overload, hindering the public's ability to efficiently access essential information. The diverse content of science and technology news encompasses multiple dimensions, including technology policy, technological development, and regional innovation, complicating traditional single-label classification methods that fail to meet practical needs. Consequently, hierarchical multi-label classification methods have emerged as effective tools for facilitating multi-dimensional thematic analysis. This approach enables nuanced differentiation among various themes within the news, extracting critical information from the dynamically evolving landscape of technology news. By conducting regular hierarchical multi-label classification of online technology news texts, researchers can systematically analyze distinct stages of technological development, capturing technological breakthroughs, policy adjustments, and market changes over various time periods. This approach holds significant academic value and practical implications for exploring pathways of technological advancement (*Wang et al., 2023*) and predicting future directions in technology development (*Liu, Fang & Qin, 2022*).

Technological innovation is a long-term evolutionary process, and science and technology news chronicles key milestones and events throughout this journey. As new technologies, policies, and discoveries continually emerge, the content of science and technology news frequently undergoes dynamic changes. To promptly identify and comprehend significant changes and trends in the field, it is essential to regularly classify and extract information from frequently published technology news. However, the short-cycle collection methods for high-frequency news frequently fail to produce large-scale datasets. Although existing methods have enhanced model performance in small sample feature extraction through techniques such as feature selection (*Dhal & Azad, 2023*), transfer learning (*Ahn et al., 2019*), and self-supervised learning (*Chen et al., 2020*), they have not fully harnessed prior knowledge within the domain. Domain knowledge can furnish additional contextual information for classification models, compensating for limited sample sizes and facilitating the accurate identification of key information within complex news themes.

Existing methods for addressing complex hierarchical multi-label classification tasks with small sample data commonly encounter two major challenges: the difficulty of fully leveraging and extracting prior domain knowledge, and the inability to effectively capture deep hierarchical structural information. To tackle these issues, this paper proposes a hierarchical multi-label classification model for technology news that is enhanced by heterogeneous graph semantics. Through a hierarchical feature propagation mechanism, low-level features can acquire beneficial information from high-level features, thereby enhancing the model's ability to capture hierarchical structures. The model utilizes the multi-dimensional representation capabilities of heterogeneous graphs in conjunction

with prior knowledge from domain knowledge graphs, enhancing the understanding of semantic and structural information within small sample data and further enriching the representation of heterogeneous graphs, thereby significantly improving classification accuracy and adaptability. The contributions of this paper are as follows:

(1) proposing a novel representation model based on heterogeneous graph semantic enhancement that integrates domain knowledge graphs to effectively augment the semantic features of news texts and enhance classification performance.

(2) employing a hierarchical propagation module to capture the hierarchical structural features of technology news, thereby revealing the hierarchical classification system of texts and facilitating in-depth analysis of complex themes.

(3) addressing initial classification scenarios in data-scarce or unknown domains, providing innovative solutions for hierarchical multi-label classification tasks, and demonstrating significant application potential through comparative experimental results.

## RELATED WORK

The classification of online text information resources has long been a focal point of research in the field of information resource organization and serialization, drawing considerable attention from both scholars and industry practitioners. This growing interest is reflected in key areas such as methods for extracting text features, the development of innovative classification models, and their practical applications across various industries.

Feature selection, enhancement, and utilization are pivotal in addressing the complexity of online text classification tasks efficiently. Significant research efforts have been devoted to this domain. For feature selection, *Dhal & Azad (2021)* introduced a method that combines particle swarm optimization (PSO) and gray wolf optimization (GWO) based on Newton's laws of motion. This method aims to achieve an optimal balance among multiple features through multi-objective optimization, advancing the performance of text classification by fine-tuning deep learning models (*Dhal & Azad, 2024*). In terms of data augmentation, *Zhao et al. (2024)* enhanced classification model performance by utilizing data augmentation and contrastive learning within heterogeneous graphs. *Wang et al. (2024)* applied back-translation for data augmentation in news texts, employing RoBERTa-RCNN with attention mechanisms to extract salient features. Additionally, *Lin et al. (2021)* proposed a heterogeneous graph structure based on textual data, leveraging pre-trained models and transductive learning to extract features. As for external feature supplementation, *Li et al. (2022)* improved the contextual feature representation capabilities of Transformer models by incorporating entities and relationships from knowledge graphs, thereby enhancing classification accuracy. *Yu et al. (2022)* classified Chinese small-sample news by incorporating external knowledge and prompt learning techniques. Furthermore, *Jang et al. (2021)* introduced a novel knowledge-injection attention mechanism that integrates higher-level concepts into neural networks to achieve both accurate and interpretable text classification. In the realm of multi-modal feature extraction, *Zhu et al. (2023)* employed multi-modal feature mining, integrating features from text, images, and audio to further improve classification performance.

In the field of innovative classification model research and industry applications, the construction and implementation of hierarchical multi-label classification models have become increasingly significant. These models systematically organize information into various themes and levels, facilitating users in quickly locating and retrieving relevant data. Hierarchical multi-label classification assigns multiple interrelated labels to each entity within a hierarchical structure, ranging from general to specific categories. *Yang et al. (2016)* proposed a Hierarchical Attention Network (HAN), which enhances classification performance by capturing the multi-layered structure of texts. *Shimura, Li & Fukumoto (2018)* designed a CNN-based approach that utilizes upper-level data to assist in lower-level classifications. *Banerjee et al. (2019)* trained a classifier for each label, introducing a parameter-sharing strategy from parent to child models. *Zhou et al. (2020)* employed a structural encoder to model label dependencies in both top-down and bottom-up directions. *Chen et al. (2021)* used BERT as an encoder and proposed a matching network to explore the relative distances between texts and labels. *Wang et al. (2022)* incorporated a hierarchical structure into the BERT encoder through contrastive learning. *Huang & Liu (2022)* developed the MSML-BERT model to tackle hierarchical multi-label text classification, embedding a multi-label classification framework within BERT to better manage complex label dependencies, demonstrating strong performance in large-scale hierarchical classification tasks. *Jiang et al. (2024)* applied conventional classification models such as TextCNN and TextRNN_Att for multi-label classification of academic papers, with attention mechanisms further improving classification performance. *Dhal et al. (2023)* investigated a stacked-layer deep learning approach for fake news classification. Table 1 presents a comparative analysis of the advantages and disadvantages of previous existing works. However, most of these methods fail to address the challenges posed by small sample datasets and the integration of relevant prior knowledge, limiting their ability to fully utilize domain-specific features and knowledge structures.

## METHODS

Science and technology news texts can be categorized into several broad thematic groups, which can be further divided into more specialized sub-themes. Each news article may encompass multiple topics simultaneously. This study proposes a classification model tailored for science and technology news articles, designed to manage hierarchical and multi-label data effectively. The proposed approach utilizes a heterogeneous graph representation to model the data. The goal of the model is to accurately analyze and leverage the hierarchical structural information inherent in the news data, enabling precise multi-label classification decisions. The architecture of the model is illustrated in Fig. 1.

The first step in this approach involves constructing a heterogeneous network that includes nodes representing science and technology news articles, as well as nodes representing individual words. This graph enables the extraction of both global text features and specific word-level features. The news article nodes leverage the BERT Hierarchical Transfer Module (BHTM) to capture hierarchical information from the news content. Additionally, the science and technology knowledge graph integrates entity nodes and

**Table 1** Comparison of advantages and disadvantages of different hierarchical multi-label classification models.

| Existing research | Advantages | Disadvantages |
|---|---|---|
| *Yang et al. (2016)* | Improved document classification performance and interpretability through hierarchical structure and dual attention mechanism. | High computational complexity and limited performance with sparse data. |
| *Shimura, Li & Fukumoto (2018)* | HFT-CNN combines convolutional neural networks with hierarchical structure, suitable for short text classification, and can learn category hierarchies. | Limited performance on long texts or datasets with significantly varying category hierarchies, insufficient generalization ability. |
| *Banerjee et al. (2019)* | Proposed a hierarchical transfer learning method that leverages existing model knowledge in new tasks to improve classification efficiency. | Transfer learning depends on the quality of the source model, high computational resource requirements, not suitable for small sample scenarios. |
| *Zhou et al. (2020)* | Introduced a hierarchy-aware global model that can simultaneously handle hierarchical label dependencies and global text features, offering strong classification robustness. | High computational resource requirements; increased information redundancy when the label hierarchy is complex, affecting efficiency. |
| *Chen et al. (2021)* | The HiMatch model enhances hierarchical classification performance through a label semantics matching network, achieving high classification accuracy by matching semantic features to label hierarchies. | Poor performance on datasets with weak label semantic relationships, and high computational resource consumption. |
| *Wang et al. (2022)* | The HGCLR model combines contrastive learning and graph structure, performing well in small sample and complex hierarchical label tasks while capturing label dependencies. | The design of positive and negative samples in contrastive learning increases the complexity of model training, revealing computational bottlenecks when handling large hierarchical structures. |
| *Huang & Liu (2022)* | The MSML-BERT model combines BERT with multi-label classification structure, suitable for complex hierarchical multi-label tasks, with good generalization ability and classification performance. | Lower efficiency when handling a large number of labels, difficulties in processing large datasets. |
| *Jiang et al. (2024)* | Optimized a general multi-label classification model by incorporating domain features of academic papers, enhancing the model's performance in specific scenarios. | The model has weak interpretability. |

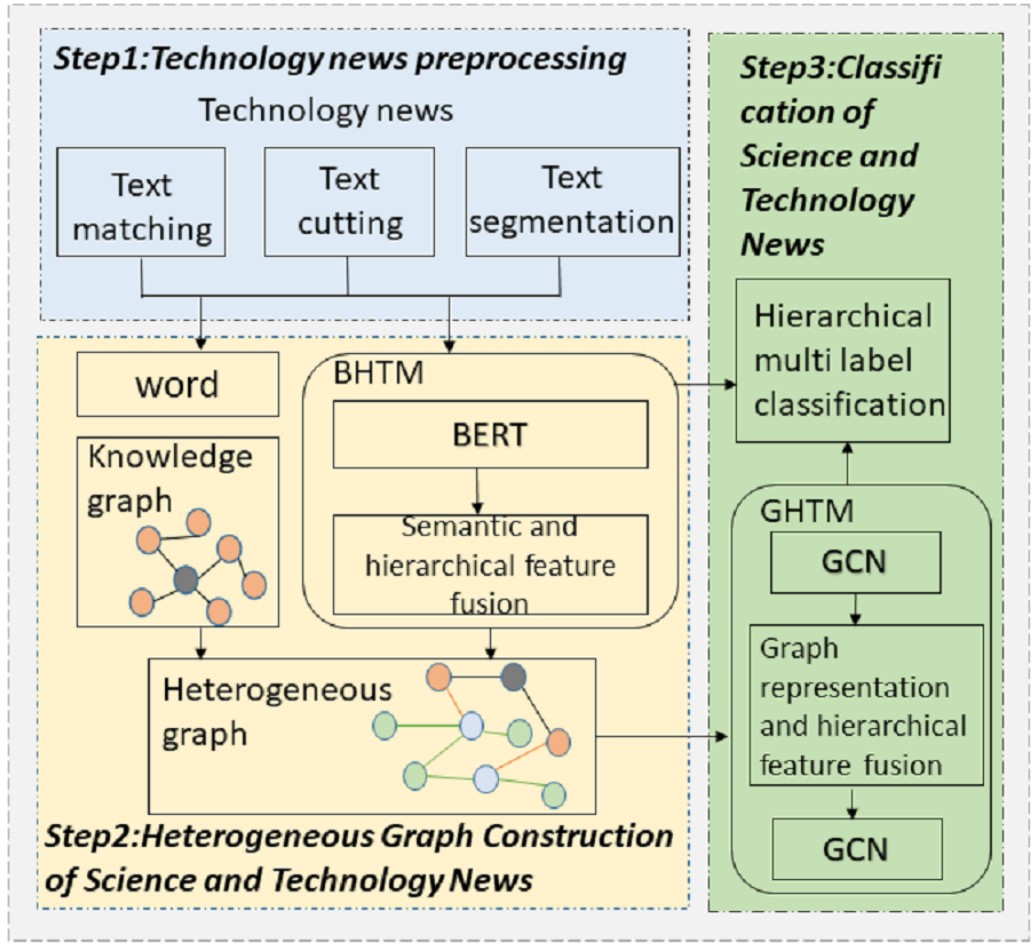

**Figure 1** **Multi label classification model for technology news text hierarchy based on heterogeneous graph semantic enhancement.**

their relationships into the heterogeneous network of science and technology news. The Graph Convolutional Neural Network Hierarchical Transfer Module (GHTM) employs a dual-layer graph convolutional network with hierarchical transmission to update node representations by combining the features of the nodes and their neighbors, along with lower-level coarse-grained topics. Word nodes, article nodes, and entity nodes are capable of integrating not only their own features and those of their neighboring nodes but also more detailed thematic information. Ultimately, the process of hierarchical multi-label classification of science and technology news is achieved using these combined features. The hierarchical multi-label classification model for science and technology news texts, enhanced by semantic information from a heterogeneous graph, consists of three key components: technology news heterogeneous graph construction module, feature fusion module based on hierarchical multi-label messaging, and hierarchical multi-label text classification module.

## Technology news heterogeneous graph construction module

Heterogeneous graphs, which represent multiple types of nodes and their relationships, enhance data representation capabilities in scenarios with limited sample sizes. By constructing a heterogeneous graph that incorporates both document and word nodes, document nodes can share word nodes and establish connections with other documents, facilitating information transfer across different nodes. This structure captures similarities and dependencies between categories, thereby improving the performance of small sample classification. However, small sample documents often fail to fully cover all label features within a specific domain, making it challenging to capture the complex dependencies among hierarchical labels based solely on textual information. In this context, knowledge graphs, as external representations of domain knowledge, provide rich prior information to assist the model in supplementing knowledge when data is insufficient. By constructing a knowledge graph centered on labels, label nodes can aggregate extensive prior knowledge from various categories and interact with corresponding word and document nodes *via* entity nodes within the heterogeneous graph, thereby enhancing the model's ability to capture the dependencies among hierarchical labels.

This paper proposes a heterogeneous graph construction strategy specifically designed to capture the unique characteristics of science and technology news texts, aiming to fully leverage both textual and entity information within these articles. The constructed heterogeneous graph comprises three types of nodes: science and technology news text nodes, word nodes, and technology entity nodes. The science and technology news text nodes are derived from the preprocessed original news articles, word nodes represent significant terms extracted from the texts through tokenization, and technology entity nodes are sourced from a custom-built technology knowledge graph, which focuses on science and technology knowledge relevant to the news topics. This approach enables the heterogeneous graph to seamlessly integrate both textual data and external knowledge, thereby enhancing the model's performance in classification and information extraction tasks related to science and technology news.

As shown in Fig. 2, this paper constructs a heterogeneous graph $G = (V, E)$ comprising various types of nodes. The node set $V$ consists of the science and technology text set $D = \{d_1, d_2, \ldots, d_m\}$, the word set $W = \{w_1, w_2, \ldots, w_n\}$, and the science and technology entity set $S = \{s_1, s_2, \ldots, s_k\}$, such that $V = D \cup W \cup S$. In this heterogeneous graph, the total number of nodes $\lceil V \rceil$ is the sum of the number of science and technology texts, science and technology-related words, and science and technology entities. The edge set $E$ consists of four distinct types of edges: associations between science and technology texts and words, co-occurrence relationships between words, associations between science and technology texts and entities, and hierarchical relationships between entities as originally defined in the knowledge graph.

A vocabulary library and a stopword library are constructed, and Jieba tokenization is used to remove irrelevant words from the science and technology news texts, while identifying and retaining the effective vocabulary set $W$ from document $D$. This process results in a vocabulary set that spans the entire corpus. When a specific term appears in a science and technology news text, an edge is established between the news text and

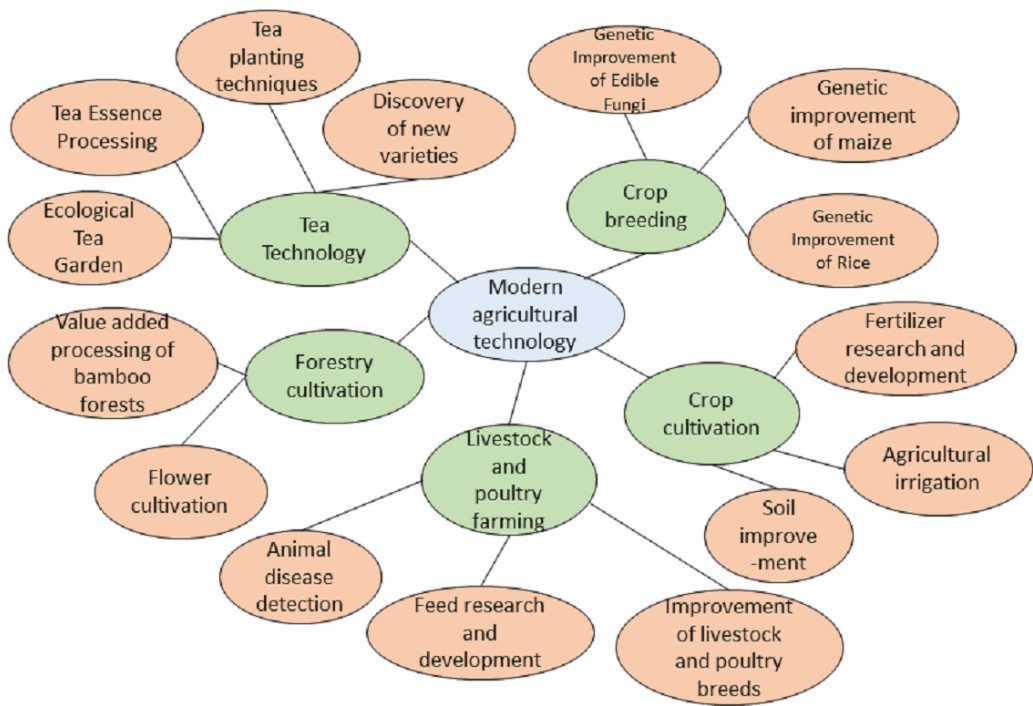

**Figure 2  Technology knowledge graph (local).**

the corresponding vocabulary. The edge weight is calculated using the term frequency-inverse document frequency (TF-IDF) metric, which aids in extracting key content from the document. To capture global co-occurrence information between words, this study employs pointwise mutual information (PMI) to quantify the co-occurrence relationships between vocabulary terms. By establishing edges between word nodes based on PMI values, the structural information of the heterogeneous graph is further enriched, enhancing the model's ability to capture associations among vocabulary terms.

The technology knowledge graph organizes information hierarchically, extending lower-level entities from higher-level entities. Higher-level entities represent broader or more abstract concepts, while lower-level entities represent more specific or detailed concepts. Using the "14th Five-Year Plan for Scientific and Technological Innovation Development in Fujian Province" as a case study, the focus is on key technology development policies, from which a technology knowledge graph is constructed (Fig. 3) to introduce prior technological knowledge for the hierarchical multi-label classification of science and technology news texts. Lower-level entities in the technology knowledge graph establish edge links with specific words in the technology news texts, while higher-level entities can be abstracted as labels for technology news. Preserving the hierarchical relationships among technology entities provides a logical foundation for multi-label classification. The hierarchical relationships among entities are stable; therefore, the maximum value of all TF-IDF scores is used as the edge weight between higher and lower-level entities. Some vocabulary in the technology news texts has semantic similarities to technology entities. By

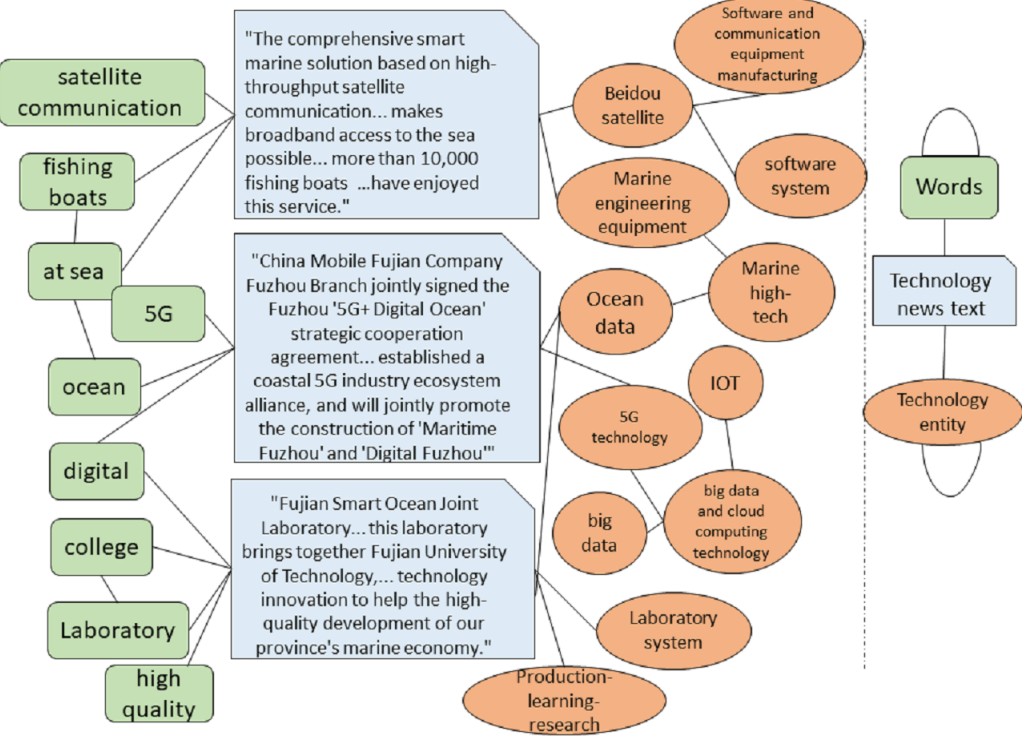

**Figure 3  Example of heterogeneous graph.**

calculating the cosine similarity between the vocabulary in the technology news texts and technology entities, an edge is established between the news text and the corresponding technology entity when the cosine similarity exceeds a predetermined threshold. The weight of these edges is determined by the TF-IDF value of the vocabulary, reflecting the degree of association between the technology news text and the technology entity. By constructing a heterogeneous graph structure, this approach innovatively combines internal textual features with external domain knowledge, offering a novel structured representation for the classification of science and technology news texts.

## Feature fusion module based on hierarchical multi-label messaging

The feature fusion module incorporates a hierarchical multi-label information transfer mechanism, consisting of BHTM and GHTM. By employing the hierarchical feature transfer method, the model efficiently recognizes and utilizes the hierarchical organization of the data to accomplish precise categorization of multiple levels of categories.

BERT offers profound and exceptionally contextualized feature representations that may extract the comprehensive characteristics of text. The graph convolutional network enhances the representation of each node by examining the interactions among nodes in the graph, where the variations in the weights of each node additionally determine the efficacy of these interactions. By constructing diverse graphs within the entire corpus for the purpose of mining features using graph convolutional networks, it becomes feasible

to develop links between various texts, capture indirect co-occurrence data, and thereby augment the complexity of text characteristics. Let X be the initial feature matrix for the nodes, which is an identity matrix. This matrix includes the total number of science and technology news text nodes ($n_{doc}$), word nodes (n_word), and science and technology entity nodes ($n_{entity}$). The feature matrix X_doc for the science and technology news text nodes is represented by the initial embedding vectors from the BERT model. The feature matrix $X_{word}$ for the word nodes is encoded using the Word2Vec word vectors, while the feature matrix $X_{entity}$ for the scientific and technological entity nodes is also encoded using the Word2Vec word vectors. This configuration facilitates a comprehensive exploration of the intricate linguistic characteristics of science and technological news texts and their contextual associations.

The BERT-based Hierarchical Multi-Label Information Transfer Layer utilizes the semantic extraction capability of BERT and the Hierarchical Multi-Label Information Transfer to provide anticipated probability distributions of category labels at various levels. The ultimate result of the probability distribution $P_h$ for predicting category labels in the second layer include both the semantic information features of the sentence context and the category judgement information from the first layer. The input features undergo processing using a multilayer perceptron that employs a rectified linear unit (ReLU) activation function to increase nonlinear expression. To minimize overfitting, a dropout mechanism is employed.

$$P_h = \left( w_P^h Relu \left( w_a^h \left( A_h \oplus D_{[CLS]} \right) + b_a^h \right) \right) + b_P^h \tag{1}$$

where weight $w_a^h$ and bias vector $b_a^h$ are the learnable parameters in the first hidden layer of MLP; weight $w_P^h$ and bias vector $b_P^h$ are the learnable parameters in the second hidden layer.

The purpose of integrating semantic feature and layer feature fusion is to mitigate the issue of feature dilution that arises from going through multiple layers. This integration ensures that the model can efficiently leverage the deep classification information. In this procedure, the first layer's output not only contributes its own category prediction results but also enhances the second layer with valuable contextual information by merging with the original characteristics. This allows the model to capture the significant interconnections between the hierarchical labels, resulting in a more precise and detailed hierarchical classification.

The hierarchical multi-label information transmission layer in the graph convolutional network (GCN) utilizes a hierarchical technique to collect and leverage the hierarchical properties of graph-structured data. The paper introduces two consecutive graph convolutional layers, each designed for a specific classification task at a different level. This technique efficiently detects and categorizes several levels of classifications while maintaining the original graph structure data.

Initially, the graph convolution operation is utilized to modify the feature representation of each node. This is achieved by gathering the feature information from neighboring nodes

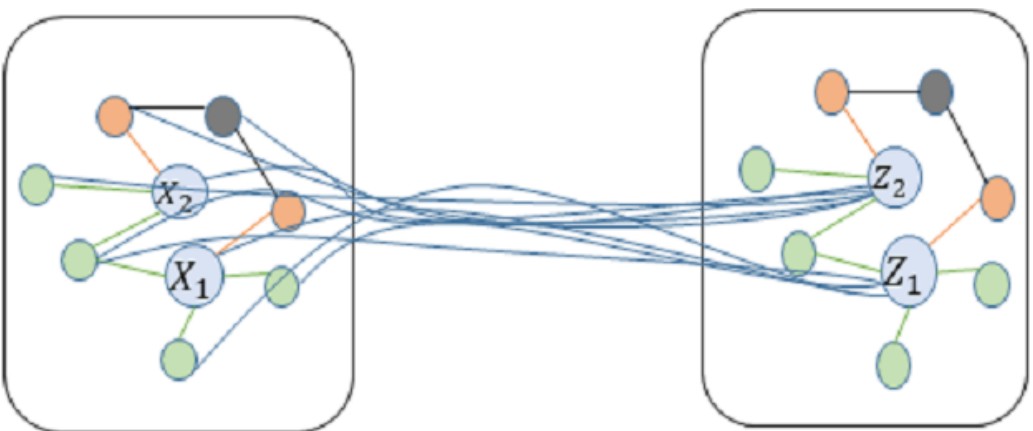

**Figure 4   Schematic diagram of convolution for technology news text image.**

using a specific formula, as depicted in Fig. 4.

$$H^{(l+1)} = \sigma\left(\tilde{D}^{-\frac{1}{2}}\tilde{A}\tilde{D}^{-\frac{1}{2}}H^{(l)}W^{(l)}\right) \tag{2}$$

$H^{(l)}$ represents the node feature matrix at the l-th layer. $\tilde{A} = A + I_N$ denotes the adjacency matrix A of the graph plus the identity matrix $I_N$,, ensuring that each node includes a self-loop, thereby considering the node's own features when aggregating neighbour information. $\tilde{D}$ is the diagonal degree matrix computed from $\tilde{A}$, where each element $\tilde{D}_{ii}$ equals the total number of edges connected to node i (including the self-loop). $W^{(l)}$ is the trainable weight matrix at the l-th layer, used to transform the feature space. $\sigma(\cdot)$ is a nonlinear activation function, enhancing the model's ability to express nonlinear relationships.

During the initial stage of hierarchical convolution, the feature $H^{(l+1)}$ of each tech news node incorporates not only its own information but also the information from neighboring word nodes and tech entities. This process results in the updated feature ZGCN for the tech news node. The initial stage of the graphical convolution classifier analyzes these feature representations to generally categorize the data into multiple overarching groups, establishing the basis for subsequent, more intricate categorization.

A cascading information transfer layer integrates the output characteristics of the initial classifier layer with the output characteristics of the final graph convolutional layer. The second layer of the GCN utilizes the fused deep features to gather and enhance the features of neighboring nodes, hence improving the accuracy of the multi-label classification. Based on the initial broad categories, the analysis delves into the adjacent node information associated with these categories. This process divides the news text into more specific subcategories, resulting in a more comprehensive and detailed understanding and division of the hierarchical structure of science and technology news data.

## Hierarchical multi-label text classification module

An essential obstacle in hierarchical multi-label text classification is the efficient representation of numerous labels for a given sample. In a classification issue where only one label is assigned to each instance, the labels can typically be represented by straightforward numerical values. Every potential combination of labels is transformed into a distinct binary vector through the process of one-hot encoding. The length of the vector corresponds to the total count of unique labels. Every element in this vector corresponds to a distinct label. If the sample possesses the designated label, the corresponding value is assigned as 1; otherwise, it is assigned as 0.

The fusion feature $Z_{GCN}$ of the graph convolutional neural network-based hierarchical multi-labeled messaging layer, along with the BERT-based hierarchical multi-labelled messaging layer, captures the contextual hierarchical feature $Z_B$ as the overall feature of the scientific and technological text Z. The feature Z is processed by a fully-connected layer with a sigmoid activation function. The number of output nodes in this layer is equal to the predetermined number of labels. This layer predicts the likelihood of each label, assuming a hierarchical multi-tag classification approach.

$$Z = \theta Z_{GCN} + (1-\theta)Z_B \tag{3}$$

$$\text{prediction} = \text{sigmoid}(Z) \tag{4}$$

In a hierarchical multi-label text classification problem, each sample can be associated with multiple labels per level, necessitating a loss function that can handle multiple positive category labels. For each sample $i$ and each category $j$, the loss function is calculated as follows:

$$L_{1|2} = -\frac{1}{N}\sum_{i=1}^{N}\sum_{j=1}^{C}\left(y_{ij} \cdot log\left(\sigma\left(x_{ij}\right)\right) + \left(1 - y_{ij}\right) \cdot log\left(1 - \sigma\left(x_{ij}\right)\right)\right) \tag{5}$$

Let N represent the number of samples and C the number of categories. $y_{ij}$ is the true label of the $i$-th sample for the $j$-th category at this level, and $x_{ij}$ is the raw output of the multi-label classification model for the $j$-th category of the i-th sample at this level. The sigmoid activation function is denoted as $\sigma$, where $\sigma(x) = \frac{1}{1+e^{-x}}$.

The loss function computes a binary cross-entropy loss for each sample and each category at all levels. The losses for all samples and categories are summed and averaged. By minimising this loss $L$, the hierarchical multi-label text classification model learns to accurately predict the presence or absence of each category.

$$L = L_1 + L_2 \tag{6}$$

## Model performance

This work focuses on doing hierarchical multi-label classification on science and technology news texts. Each text sample has the potential to be assigned to numerous categories simultaneously. Assessing the hierarchical multi-label classification model usually requires the use of a range of metrics to thoroughly evaluate the model's performance, such as

**Table 2  Conceptual table of positive and negative examples.**

| Prediction category/ Real category | Positive example | Negative example |
|---|---|---|
| Positive example | TP | FN |
| Negative example | FP | TN |

precision, recall, and F1 scores. These metrics evaluate the model's capacity to accurately identify all pertinent labels by taking into account all the labels associated with each text sample.

In multi-label class, the "micro" averaging strategy for calculating precision involves the combined aggregation of results across all categories, followed by a measure of precision based on the overall number of true positives (TPs) *versus* false positives (FPs) obtained from the aggregation, as shown in Table 2.

$$Precision_{micro} = \frac{\sum_i TP_i}{\sum_i TP_i + \sum_i FP_i}. \tag{7}$$

Recall is determined based on an overall aggregation of the number of true positives (TPs) and false negatives (FNs) across all categories, and then calculated based on these aggregated values.

$$Recall_{micro} = \frac{\sum_i TP_i}{\sum_i TP_i + \sum_i FN_i}. \tag{8}$$

The F1 score is an evaluation metric that considers both precision and recall.

$$F1_{micro} = 2 \times \frac{Precision_{micro} \times Recall_{micro}}{Precision_{micro} + Recall_{micro}} \tag{9}$$

where TP represents the frequency of positive samples being classified as positive, FN represents the frequency of positive samples being classified as negative, FP represents the frequency of negative samples being classified as positive, and TN represents the frequency of negative samples being classified as negative.

To better understand the computational resource requirements of our proposed model, the model's time complexity primarily depends on the construction of the heterogeneous graph, global feature extraction using BERT, and convolutional node feature extraction *via* GCN. The time complexity for constructing nodes is O($N_d + N_w + N_e$). where $N_d$ is the total number of documents, $N_w$ isthe total number of words, and $N_e$ is the total number of entities. The edge construction, based on the relationships between words, documents, and entities, has a time complexity of approximately O($N_d N_w + N_w N_w + N_e N_w$). The time complexity for BERT's global feature extraction is O($n^2 d_1$), where n s the length of the input sequence and $d_1$ is the hidden layer dimension of the BERT model. GCN's convolutional node feature extraction has a complexity of O($V d_2 + E$), where $V$ is the total count of documents, words, and entities, $E$ is the number of edges in the graph, and $d_2$ is the feature dimension of the nodes. Therefore, the overall complexity of the model is expressed as: O($N_d + N_w + N_e + N_d N_w + N_w N_w + N_e N_w + n^2 d_1 + V d_2 + E$).

**Table 3  Technological innovation system.**

| First level | Second level |
| --- | --- |
| Knowledge Innovation | Talent Innovation |
| | Industry-University-Research Integration |
| | Competition Conferences |
| Technological Innovation | Green and Low-Carbon Technologies |
| | Internet of Things (IoT) Technologies |
| | High-Tech Marine Technologies |
| | Modern Agricultural Technologies |
| | Public Service Technologies |
| | Integrated Circuits and Key Components |
| | New Material Technologies |
| | Artificial Intelligence (AI) Technologies |
| | Life and Health Technologies |
| | Advanced Manufacturing Technology |
| | New Energy Technologies |
| | Software and Communication Equipment Manufacturing |
| Management Innovation | Intellectual Property Management |
| | Science and Technology Finance |
| | Tax Support |
| | Technological Assistance |
| Regional Innovation | Science and Technology Corridors |
| | High-Tech Zones |
| Innovation Platforms | Laboratory Systems |
| | Research Institutes (Centers) |
| | R&D Institutions |

# RESULTS AND ANALYSIS

## Classification system construction and results

Existing research suggests that the science and technology innovation system is a complex entity comprising three subsystems: knowledge innovation, technological innovation, and management innovation, all driven by modern technology. These subsystems not only interpenetrate and support one another but also collectively drive the advancement of technological innovation. At the same time, the construction of innovation platforms and regional innovation develops through a multifaceted interplay of knowledge, technology, and management from various perspectives. Table 3 presents a detailed overview of the key components of the science and technology innovation system and their hierarchical structure. The hierarchical nature and diversity of the science and technology innovation system closely correspond to the complexity of science and technology news content, thereby providing an effective framework for guiding the multi-label hierarchical classification of such news.

By utilizing web crawler technology, we captured science and technology news reports from the "Southeast Network". The individual responsible for overseeing the science and technology division often introduces these reports, and the associated scientific and

**Table 4  Technology news text annotation data.**

| Text | Category |
| --- | --- |
| "The comprehensive smart marine solution based on high-throughput satellite communication... makes broadband access to the sea possible... more than 10,000 fishing boats nationwide have enjoyed this service." | Software and Communication Equipment Manufacturing, High-Tech Marine Technologies |
| "Ningde implements the energy storage industry... breakthrough in advanced energy storage material research and development... also established the country's first national intellectual property protection center specifically serving the power battery industry cluster..." | New Material Technologies, Intellectual Property |
| "China Mobile Fujian Company Fuzhou Branch jointly signed the Fuzhou '5G+ Digital Ocean' strategic cooperation agreement... established a coastal 5G industry ecosystem alliance, and will jointly promote the construction of 'Maritime Fuzhou' and 'Digital Fuzhou'" | High-Tech Marine Technologies, Internet of Things (IoT), Big Data and Cloud Computing Technologies |
| "Fujian Smart Ocean Joint Laboratory... this laboratory brings together Fujian University of Technology, Jimei University... will cooperate in the field of marine science and technology innovation to help the high-quality development of our province's marine economy." | High-Tech Marine Technologies, Industry-University-Research Integration, Laboratory Systems |

technological material is expounded upon in great depth. In order to efficiently retrieve important information from these stories, we utilized a keyword targeting approach to identify terms like "introduction" and "understanding", thus precisely capturing significant content from the science and technology news. This solution circumvents the conventional, laborious procedure of capturing information by means of a lengthy text sliding window. It not only efficiently preserves the essential content of science and technology news but also significantly minimizes the inclusion of irrelevant distractions. We performed word segmentation on the intercepted science and technology news material using a predefined, restricted, and deactivated thesaurus. Afterwards, we manually assigned multiple labels to the text. Table 4 displays instances of data that have several labels assigned at the second level.

The processing flow of the model is depicted in Fig. 5. First, the text is tokenized, and a heterogeneous graph is constructed that links documents to words. While terms related to "agriculture" may not explicitly appear in the document, potentially hindering traditional models from extracting sufficient features for accurate classification, the proposed model leverages prior knowledge from the technology domain knowledge graph. For instance, agricultural technology-related entities such as "soilless cultivation" and "greenhouse" enable the word node "soilless cultivation" to connect with entity nodes in the agricultural sector. By using a graph convolutional network to explore the relationships between nodes in the heterogeneous graph and employing a hierarchical feature propagation mechanism, higher-level technical domain knowledge is transmitted to lower-level features, providing

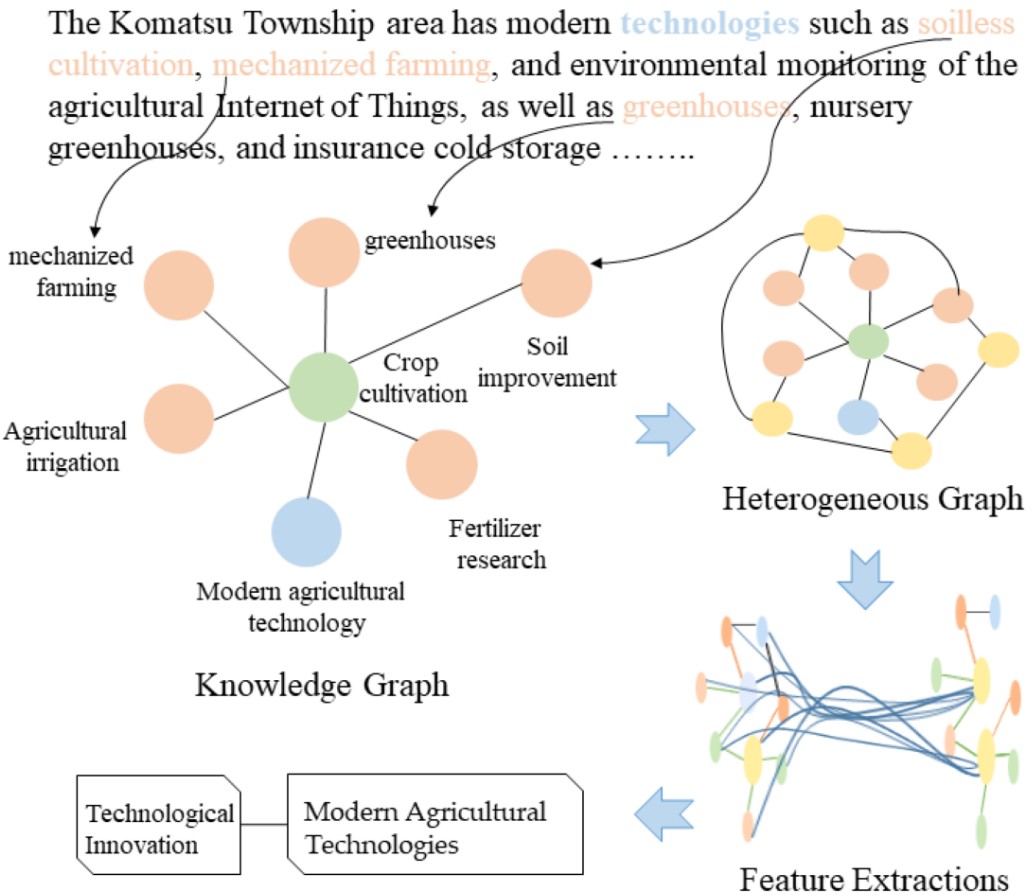

**Figure 5** Demonstration dataset integration and model execution.

additional semantic context for the document. This approach ultimately helps the model classify the document accurately under the category "Technological Innovation: Modern Agricultural Technologies."

In the overall evaluation of the dataset, the model achieved precision, recall, and F1 scores of 84.21%, 88.89%, and 86.49%, respectively. These results demonstrate a high level of classification performance, largely attributable to enhancements in both the model architecture and the handling of hierarchical label relationships. First, the integration of a heterogeneous graph representation with a domain knowledge graph enables a more refined capture of the relationships between science and technology news texts and hierarchical labels. By incorporating both text nodes and domain-specific entity nodes, the model significantly improves its understanding of text semantics, particularly in data-scarce scenarios, where it performs remarkably well despite limited training data. Second, the model captures the relationships between coarse-grained and fine-grained labels through a hierarchical label propagation module. During the classification process, the model considers not only local features (such as word-level information) but also

**Table 5  Performance comparison with benchmark classification models.**

| Model | Precision/% | Recall/% | F1 /% |
|---|---|---|---|
| TextRNN | 79.58 | 45.95 | 57.01 |
| FastText | 78.75 | 45.11 | 57.36 |
| TextCNN | 85.62 | 64.11 | 73.32 |
| BERT | 89.43 | 70.81 | 79.04 |
| RoBERTa | 87.11 | 75.00 | 80.60 |
| This model | 84.21 | 88.89 | 86.49 |

broader contextual information (such as document-level context), leading to more precise hierarchical multi-label classification.

## Model comparison

To evaluate the performance of the hierarchical multi-label classification model for science and technology news text based on heterogeneous graph representation presented in this paper, five popular benchmark models were used for comparative experiments. These comparison models employ the same hierarchical delivery method as the model in this study. TextRNN (*Liu, Qiu & Huang, 2016*): a text classification model based on recurrent neural networks (RNN), utilizing the sequence processing capability of RNNs to capture temporal information in text. FastText (*Joulin et al., 2017*): a model based on a simplified linear architecture that uses bag-of-words and N-gram features to quickly train and classify text. TextCNN (*Kim, 2014*): a model that utilizes convolutional neural networks (CNNs) to capture and classify local features in the text using convolutional kernels of various sizes. BERT (*Devlin et al., 2018*): a pre-trained deep learning model based on the Transformer architecture that learns bi-directional contextual representations from a large-scale text corpus. RoBERTa (*Liu et al., 2019*): this model utilizes a larger dataset and an unsupervised pre-training approach, employing a more dynamic masking mechanism, which enhances its performance in natural language processing tasks. Table 5 presents the precision, recall, and F1 scores of second-level multi-label classification on technology news text using this model in comparison with the aforementioned benchmark models.

From the comparison results in Table 5, it is evident that the proposed model demonstrates significant improvements in recall and F1 score compared to all baseline models. Specifically, the recall rate increased by 13.89% and the average F1 score improved by 6.11% compared to the best-performing RoBERTa model.

To further validate the advantages of our model in capturing hierarchical structure information and enhancing classification performance, we compared it with three models that have demonstrated strong performance in hierarchical multi-label classification tasks in recent years: HAN (*Yang et al., 2016*), HiMatch (*Chen et al., 2021*), and HGCLR (*Wang et al., 2022*). The HAN model captures both global and local features of the text through a hierarchical attention mechanism, effectively extracting hierarchical information by utilizing sentence-level and word-level attention in hierarchical text classification tasks. The HiMatch model improves the understanding of label hierarchy by combining the semantic information from both text features and labels. HGCLR models the hierarchical

**Table 6  Performance comparison with advanced hierarchical multi-label classification models.**

| Model | Precision/% | Recall/% | F1 /% | Time/min |
| --- | --- | --- | --- | --- |
| HAN | 81.47 | 74.34 | 77.60 | 79.8 |
| HiMatch | 82.29 | 80.56 | 81.42 | 77.9 |
| HGCLR | 83.78 | 86.11 | 84.93 | 98.6 |
| This model | 84.21 | 88.89 | 86.49 | 102.5 |

relationships between text and labels as a graph structure and employs a GCN to capture these relationships, enhancing the model's ability to capture label hierarchy through contrastive learning.

The proposed model incorporates a heterogeneous graph construction mechanism that integrates prior knowledge to enhance the understanding and modeling of hierarchical labels. Additionally, it effectively captures hierarchical dependencies between labels through a dual-layer BERT and GCN, resulting in exceptional performance in hierarchical label classification tasks. These design elements contribute to precision (84.21%) and recall (88.89%) rates that outperform those of HAN, HiMatch, and HGCLR, as shown in Table 6. However, the combination of complex mechanisms—particularly the integration of heterogeneous graph construction with prior knowledge, the BERT pre-training model, and GCN—significantly increases computational overhead. Each module requires extensive parameter updates and computations, leading to noticeably longer run times compared to other models. Despite this, accurately capturing label relationships is crucial in hierarchical multi-label tasks, and the model achieves high-precision classification performance through these intricate mechanisms.

## Ablation experiments

To quantify the impact of node feature embeddings in the hierarchical multi-label classification model for science and technology news text based on heterogeneous graph representation constructed in this paper, an ablation experiment comparing the performance of model components was designed to verify the model's effectiveness. Figure 6 displays the experimental outcomes.

Removing the BERT module substantially diminished the model's capacity to extract global textual features. As a pre-trained language model, BERT excels at capturing complex contextual and semantic information within the text. Its removal resulted in a notable decline in both precision and recall, with recall being particularly affected, indicating a reduced capacity to identify complete labels, which subsequently lowered the F1 score.

GCN strengthens connections between nodes through feature propagation and the aggregation of neighborhood information, offering distinct advantages in capturing multi-layered features. After the removal of GCN, recall dropped significantly from 88.89% to 70.81%, indicating a diminished capability to effectively capture global label relationships, particularly in recognizing complex label interactions and multi-layered information. However, precision increased slightly (from 84.21% to 89.43%), likely due to a more conservative classification strategy that reduced false positives, though overall performance declined, with the F1 score decreasing to 79.04%.

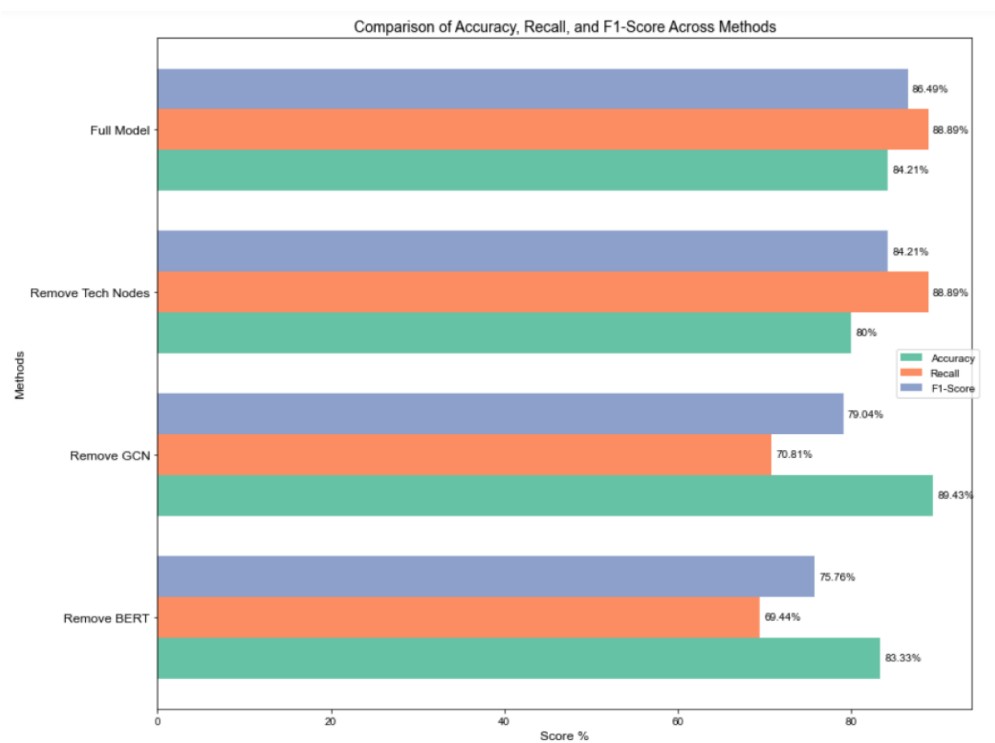

**Figure 6** Ablation experiment for comparing the performance of model components.

Removing technology entities from the knowledge graph reduced the model's precision from 84.21% to 80%, highlighting the critical role of domain-specific knowledge in improving accuracy. Domain-specific knowledge aids the model in accurately understanding text semantics and making precise classifications in complex or specialized news contexts. Without this module, the model performed poorly in classifying domain-specific labels, leading to a decrease in precision.

## Hierarchical experiments

To quantify the effectiveness of the hierarchical classification constructed in this study, a single-layer hierarchical classification algorithm, which does not consider hierarchical structural relationships, was designed. Without modifying any other variables, the No_GHTM model eliminates the Graph Convolutional Hierarchical Transmission Module and replaces it with a single-layer graph convolution. The No_BHTM model eliminates the BERT Hierarchical Transmission Module and use a single-layer BERT for extracting features. The No_Both model eliminates both the Graph Convolutional and BERT Hierarchical Transmission Modules, and instead use a single-layer BERT and graph convolutional neural network for updating node features. The conclusive categorization outcomes are displayed in Fig. 7.

The classification performance of No_BHTM declines significantly, indicating that ignoring the BERT hierarchical transmission module leads to a lack of coarse-grained labels between levels, which in turn impacts the global semantic feature representation under
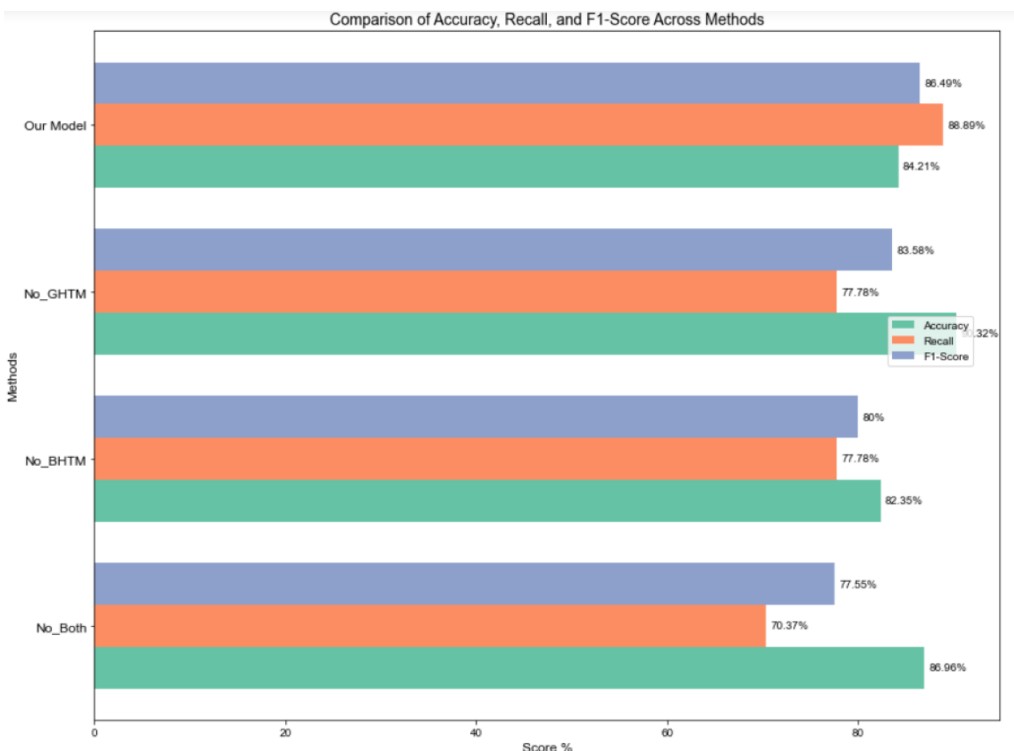

**Figure 7** Experimental results of transfer modules at different levels.

fine-grained labels. While No_GHTM, which leverages a single-layer graph convolutional network along with the BERT hierarchical transmission module for feature extraction, improves precision, its average recall and F1 score have notably decreased. This indicates that No_GHTM sacrifices recall, implying that certain true positive samples are missed by the model, resulting in a lower recall. In other words, disregarding the impact of coarse-grained hierarchical labels on the graph nodes to be classified reduces the recognition rate of positive samples. No_Both, which disregards both hierarchical transmission modules and directly predicts at the second level, exhibits a sharp decline in recall, reflecting difficulty in capturing more potential labels. The decline in F1 score further highlights the model's inability to balance precision and recall without the hierarchical transmission mechanism. This underscores the critical role that both BHTM and GHTM play in multi-level feature extraction and label propagation within the model.

## Robustness

To assess the robustness of the proposed hierarchical multi-label classification model on small-sample datasets, this study selected the RCV1-v2 dataset for experiments and processed it to construct a small-sample version. RCV1-v2 is a text classification dataset derived from Reuters news articles, comprising over 800,000 documents that span a wide array of topics. The dataset features a well-defined hierarchical label structure, organized

**Table 7** Experimental comparison results of RCV1-v2 rehierarchical multi-label text categorization model for small sample dataset.

| Model | Precision/% | Recall/% | F1 /% |
|---|---|---|---|
| HAN | 68.75 | 50.00 | 57.89 |
| HiMatch | 72.15 | 54.72 | 62.37 |
| HGCLR | 71.79 | 59.57 | 65.12 |
| This model | 73.17 | 61.22 | 66.67 |

into multiple levels, which are further refined into specific subcategories. Thus, this dataset is highly appropriate for validating hierarchical multi-label classification tasks.

To simulate a small-sample scenario, the labels in the RCV1-v2 dataset were first aggregated to the second level, with all second-level labels treated as top-level labels, thereby constructing a simplified label system. Subsequently, samples were randomly selected based on the proportion of label categories until the total sample size reached 3,000. This sampling strategy was designed to simulate small-sample scenarios commonly encountered in practical applications, to evaluate the model's generalization ability and robustness under limited data conditions.

On the processed RCV1-v2 small-sample dataset, the proposed hierarchical multi-label classification model achieved satisfactory results, with precision, recall, and F1 scores of 73.17%, 61.22%, and 66.67%, respectively, as shown in Table 7. The model's performance in the hierarchical multi-label classification task remained stable, demonstrating its robustness and adaptability in addressing small-sample challenges. These results underscore the potential of the proposed model in addressing small-sample classification tasks. Even with limited data resources, the model demonstrates the ability to maintain strong classification performance by effectively learning hierarchical label dependencies, offering critical technical support for hierarchical multi-label classification tasks in small-sample practical applications.

### Statistical test

A statistical test, specifically the two-tailed $t$-test, was conducted to demonstrate that the improvements of the proposed technique are statistically significant. Table 8 presents the $t$-values calculated from the two-tailed $t$-test, comparing the average F1 scores of the proposed method with those of other methods. For a significance level of 0.05, the $p$-values derived from the $t$-values indicate that the proposed model significantly enhances the effectiveness of classification tasks compared to other hierarchical multi-label classification models. This enhancement is attributed to its richer semantic feature extraction based on the prior knowledge graph.

## CONCLUSIONS

This study introduce a hierarchical multi-label classification model for science news texts, utilizing heterogeneous graph-based semantic enhancement. The heterogeneous graph is constructed using science news text nodes, word nodes, and science entity nodes from a knowledge graph, addressing the limitations of feature extraction in small-sample data.

**Table 8  Comparison table of statistical significance test results for different models.**

| Model | $t$-value | $p$-value |
|---|---|---|
| TextRNN | 266.46 | 1.49E−28 |
| FastText | 342.35 | 1.90E−35 |
| TextCNN | 119.04 | 2.13E−23 |
| HAN | 91.26 | 1.40E−23 |
| BERT | 76.48 | 2.51E−22 |
| RoBERT | 69.22 | 4.22E−23 |
| HiMatch | 37.26 | 2.99E−17 |
| HGCLR | 18.33 | 5.28E−13 |

The model integrates hierarchical structures and prior knowledge from domain-specific knowledge graphs, enhancing both the semantic understanding of science news and classification performance, particularly in data-scarce scenarios. The feature fusion module, driven by hierarchical multi-label information transfer, enables text classification by simultaneously accounting for both contextual and local word features, thereby facilitating the exploration of relationships between coarse-grained and fine-grained labels and resulting in refined hierarchical classification. The model exhibits strong performance in handling the complexity of multi-dimensional themes in science news, achieving high classification accuracy and surpassing traditional methods in precision, recall, and F1 score, thereby alleviating the burden of information processing.

While the proposed model demonstrates strong performance in small-sample classification tasks, its limitations should not be overlooked. The model depends on the completeness and quality of the domain knowledge graph; if the knowledge graph is incomplete or outdated, it may negatively impact classification accuracy and effectiveness. Furthermore, the model's computational complexity is relatively high, which may result in substantial consumption of computational resources and time, potentially limiting its practicality. Future research will aim to optimize the model's computational efficiency, ensuring performance is maintained while reducing resource consumption. Additionally, we will explore the model's potential applications in other fields and in zero-shot learning, particularly for addressing cold-start issues in emerging domains or data-scarce environments. By leveraging existing domain knowledge graphs, the model can facilitate faster classification and market responsiveness, thereby driving innovation in intelligent information processing technologies to meet the growing demand for efficient and accurate information classification across various fields.

### Funding

This work was supported by the Basic Research Special Project for Public Welfare Research Institutes in Fujian Province, titled "Research on the Path and Key Technologies of Science and Technology Big Data Analysis Based on News Media: A Case Study of Fujian Province"

(Project No. 2023R1008001). The funders had no role in study design, data collection and analysis, decision to publish, or preparation of the manuscript.

### Grant Disclosures

The following grant information was disclosed by the authors:

Research on the Path and Key Technologies of Science and Technology Big Data Analysis Based on News Media: A Case Study of Fujian Province: 2023R1008001.

### Competing Interests

The authors declare there are no competing interests.

### Author Contributions

- Quan Cheng conceived and designed the experiments, analyzed the data, authored or reviewed drafts of the article, and approved the final draft.
- Jingyi Cheng conceived and designed the experiments, performed the experiments, analyzed the data, performed the computation work, prepared figures and/or tables, authored or reviewed drafts of the article, and approved the final draft.
- Jian Chen conceived and designed the experiments, authored or reviewed drafts of the article, and approved the final draft.
- Shaojun Liu conceived and designed the experiments, authored or reviewed drafts of the article, and approved the final draft.

### Data Availability

The data and code are available in the Supplementary Files.

### Supplemental Information

Supplemental information for this article can be found online at http://dx.doi.org/10.7717/peerj-cs.2469#supplemental-information.

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
