# Peer review of "Hierarchical multi-label classification model for science and technology news based on heterogeneous graph semantic enhancement"

_PeerJ Computer Science, doi:10.7717/peerj-cs.2469_

## Round 0.1 · original submission · Major Revisions

Please carefully consider and revise the article according to the comments. Then the revised version will be evaluated again.

Reviewer 1 ·

Basic reporting

The following points need to be addressed for acceptance:

1. Considering the acronyms used in the article, a section on nomenclature will be helpful.
2. Does the proposed method solve or alleviate the issues faced by previous methods? How was it resolved? Maybe write more details about these parts.
3. Split the related work into the various sections.
4. Add the tabular advantages and disadvantages of the previous existing works.
5. Perform statistical tests like p-test, t-test, etc.
6. Write down the motivation of this research.
7. Compare your work with the other transformer based method
8. Used more similar datasets for the robustness of your approach.
9. Write down the contributions and problem statements of your works.
10. Add the time comparison table for your vs the existing algorithms
11. The manuscript lists multiple processing steps in detail. It is recommended to focus on the innovative part of this article and briefly summarise the existing algorithms.
12. The author needs to show the sense of the proposed method by comparison with other recent methods.
13. Add the time complexity of your method.
14. Add the learning curves for accuracy, ROC-AUC.
15. Give a reason for the results and a deeper explanation.
16. Please highlight the advantages, disadvantages, and future research of your method.
17. Mention how your model benefits the real-life problems and the tiny industries.
18. Add a demo dataset and perform the entire proposed method for reader visibility.
19. The overall quality of the paper is to be improved by going through the works like A multi-objective feature selection method using Newton’s law based PSO with GWO, A lightweight filter-based feature selection approach for multi-label text classification, Hybrid momentum accelerated bat algorithm with GWO based optimization approach for spam classification, A fine-tuning deep learning with multi-objective-based feature selection approach for the classification of text, A deep learning and multi-objective PSO with GWO based feature selection approach for text classification, Stacked Layer Based Deep Learning Approach for Fake News Classification.

Experimental design

Already added in the Basic reporting section

Validity of the findings

Already added in the Basic reporting section

Additional comments

Already added in the Basic reporting section

Reviewer 2 ·

Basic reporting

1. The English description of the paper needs to be polished. The format of the paper deserves careful revision.
2. A lot of literature is missing. The comparison method is old and not from recent years. Many of the references are not cited.
3. The picture is very unclear. The drawing of Tables 5 and 6 does not reflect the advantages of the method. It is recommended to replace the horizontal and vertical coordinates, compare each indicator together, and replace the legend with the method.
4. The motivation of the article design is not clearly stated.

Experimental design

1. The experimental comparison method is old, and the comparison results are not sufficiently reliable. Therefore, the strength of the pages reflected in the subsequent analysis cannot be recognized. It is suggested that the latest comparison method should be added.
2. Authors should write summaries carefully. Highlight research motivation and design ideas.

Validity of the findings

Compared with the old method, the results show the effectiveness of the designed method is weak.

---

## Round 0.2 · accepted · Accept

Thanks to the authors for their efforts to improve the work. The previous reviewers did not accept the invitation to review this version due to some reason. Having evaluated the revisions myself, to the best of my knowledge, I believe the authors have revised the paper appropriately. This version reaches the criteria of the journal. It may be accepted. Congrats!